# Enhancement of Ultrasonic Transducer Bandwidth by Acoustic Impedance Gradient Matching Layer

**DOI:** 10.3390/s22208025

**Published:** 2022-10-20

**Authors:** Ke Zhu, Jinpeng Ma, Xudong Qi, Bingzhong Shen, Yang Liu, Enwei Sun, Rui Zhang

**Affiliations:** 1Functional Materials and Acousto-Optic Instruments Institute, School of Physics, Harbin Institute of Technology, Harbin 150080, China; 2Functional Materials and Acousto-Optic Instruments Institute, School of Instrumentation Science and Engineering, Harbin Institute of Technology, Harbin 150080, China; 3Key Laboratory for Photonic and Electronic Bandgap Materials, Ministry of Education, School of Physics and Electronic Engineering, Harbin Normal University, Harbin 150025, China

**Keywords:** broadband ultrasound transducer, gradient acoustic impedance, matching layer, single crystal

## Abstract

High-performance broadband ultrasound transducers provide superior imaging quality in biomedical ultrasound imaging. However, a matching design that perfectly transmits the acoustic energy between the active piezoelectric element and the target medium over the operating spectrum is still lacking. In this work, an anisotropic gradient acoustic impedance composite material as the matching layer of an ultrasonic transducer was designed and fabricated; it is a non-uniform material with the continuous decline of acoustic impedance along the direction of ultrasonic propagation in a sub-wavelength range. This material provides a broadband window for ultrasonic propagation in a wide frequency range and achieves almost perfect sound energy transfer efficiency from the piezoelectric material to the target medium. Nano tungsten particles and epoxy resin were selected as filling and basic materials, respectively. Along the direction of ultrasonic propagation, the proportion of tungsten powder was carefully controlled to decrease gradually, following the natural exponential form in a very narrow thickness range. Using this new material as a matching layer with high-performance single crystals, the −6 dB bandwidth of the PMN-PT ultrasonic transducer could reach over 170%, and the insertion loss was only −20.3 dB. The transducer achieved a temporal signal close to a single wavelength, thus there is the potential to dramatically improve the resolution and imaging quality of the biomedical ultrasound imaging system.

## 1. Introduction

Ultrasonic transducers (UT), which are widely used in agriculture, and industry, especially in the field of ultrasound medical diagnosis [1,2,3,4,5,6], are mainly composed of a focusing element, a matching layer, a sound producing material and a backing block [3,4,5,6]. Among them, the matching layer is a kind of sound transmitting material located in the front of the medical ultrasonic transducer, whose main function is to efficiently channel the sound energy flow into the detection medium [4,5,6,7,8,9,10]. Because of the serious acoustic impedance mismatch between the piezoelectric element and the detection of the medium, an appropriate matching layer can transmit the sound energy efficiently between the piezoelectric chip and human soft tissue, improve the sensitivity of the transducer, broaden its frequency bandwidth and reduce the distortion [11,12].

The traditional medical ultrasonic transducers usually use PZT piezoelectric ceramics as the active material, and their matching layers are usually single-layer and double-layer, rarely more than three layers [11,12,13,14,15]. The main reason is although the matching of more than two layers can improve the overall bandwidth and detection sensitivity of the transducer in theory, the technical difficulties highlight the problems of poor bonding, porosity defects and dust impurities between multiple interfaces, which largely offset the limited improvement brought by multi-layer matching [13,14,15]. At present, the intrinsic bandwidth of single or double matching layers made by relevant research institutes and commercial institutions reach around 80%, which can only meet the requirements of traditional piezoelectric ceramic materials as the sound producing materials of ultrasonic transducers [16,17]. In order to improve the performance of advanced piezoelectric single crystal materials, better display, high efficiency, and broadband (>120%) conduction transparent (matching) materials are essential [18,19,20,21,22,23,24,25].

The acoustic impedance gradient material has been proved to exhibit the characteristics of high-frequency conduction, and the quasi-continuous acoustic impedance change can be realized in the range of material by selecting the type of filler and adjusting the filling proportion [26,27]. Therefore, using an artificial acoustic impedance gradient composite is a promising way to solve the above problem [27,28]. In recent years, many researchers have tried to apply impedance gradient composites to ultrasonic transducers with traditional piezoelectric ceramics. In 2013, Feng and Liu created a new type of micro piezoelectric ultrasonic transducer, and a gradient matching layer was fabricated by using poly (p-xylene), which improved the transmission rate of acoustic energy and the bandwidth of the transducer [27]. In 2016, Lu prepared a new anisotropic conical ultrasonic matching layer material by etching the stripped silica fiber bundle in a hydrofluoric acid solution. The corresponding −6 dB bandwidth of the ultrasonic transducer equipped with this matching layer can reach 107%, which strongly proves that the acoustic impedance gradient matching material is suitable for the broadband sound transmission ability of a single crystal^28^. However, the acoustic impedance of this structure can only change linearly along the thickness direction, and because the acoustic impedance gradient is realized by using the change in the geometric size of the high impedance filler in the matching layer, this structure is only suitable for the frequency band where the wavelength is much larger than the chassis size of high impedance filler of each unit [28].

At present, the optimal acoustic impedance gradient distribution of the matching layer is still not clear; it requires a volume ratio of high-impedance filler powder to gradually decrease along the direction of the acoustic wave propagation and achieve a large and smooth acoustic impedance span in a very thin thickness range [29,30,31,32,33,34,35]. In this paper, we confirmed the ideal acoustic impedance distribution mode and the thickness of the matching layer by simulation and realized this distribution by using nano tungsten powder with high acoustic impedance and epoxy resin to create a kind of anisotropic inhomogeneous material. Along the direction of ultrasonic propagation, the filling proportion of tungsten powder was carefully controlled to decrease gradually. Transducers with this acoustic impedance gradient matching layer were fabricated. The temporal signals and frequency spectra were tested, and their imaging resolutions were also characterized with the help of a wire phantom. 

## 2. Exploration of the Acoustic Impedance Distribution Curve of Optimal Matching Layer

In order to obtain excellent performance specifications, 1-3 piezoelectric composites with higher electromechanical coupling coefficients and relatively low acoustic impedance are employed to fabricate transducers [36,37,38] Two kinds of 1-3 composite piezoelectric materials, PZT-5H ceramic (Baoding Hongsheng Acoustical Electronics Inc., Cangzhou, China) and PMN-30PT single crystal (Shanghai Institute of Ceramics), were selected for comparison. As 1-3 PZT-5H and 1-3 PMN-PT have the same structure (same acoustic impedance and density), the simulation result works for both of them. The specific parameters are shown in Table 1.

Suitable models and equations do not exist to directly calculate the effect of matching layers with different continuous acoustic impedance distributions, so the continuous matching layer is decomposed into 10 single layers by PiezoCAD software (Sonic Concepts) to facilitate simulation. Obviously, the more layers we break down, the closer we get to the real situation. However, the simulation results in Figure 1a reveal that the choice of 10 layers is basically enough to satisfy the actual continuous situation. It can be seen that for both 1-3 P5H and 1-3 PMN-PT materials, the increase in the matching layer is conducive to the increase in bandwidth. When the number comes to 10, the increase in bandwidth is no longer obvious, and its value is close to the limit. This indicated that the simulation of 10 single layers is close to the matching layer with continuous acoustic impedance changes.

As an example, in Figure 1b, *f*_1_ and *f*_2_ refer to the lower and upper frequencies when the spectra intensity declines by 6 dB from its maximum value. The −6 dB bandwidth (BW) of the transducer can be calculated as: $$BW=\frac{2(f_{2}-f_{1})}{f_{1}+f_{2}}\times 100\%$$

There could be innumerable distribution modes of the acoustic impedance in a gradient matching layer, in which the real “ideal” one can provide the largest bandwidth for the transducer. Some examples are shown in Figure 2a. These different matching layers, together with 1-3 composite piezoelectric materials and 10.5 MRayls backing, were employed to integrate a complete transducer. The simulation results of the spectral and temporal signals were obtained with the PiezoCAD software, as shown in Figure 1b. The −6 dB bandwidth of the transducer based on all modes reached more than 100%, and the exponential acoustic impedance distribution mode, represented by red (type 4), obtained the largest bandwidth (133.6% for 1-3 PZT and 164.3% for 1-3 PMN-PT), which proves that an ideal exponential distribution of acoustic impedance matching layer will significantly optimize the ultrasonic transducer bandwidth. The final desired impedance distribution follows the law: $Z\left(x\right)=8.2e^{-4.7x}$, where *Z(x)* refers to the acoustic impedance (MRayls) at position *x* (mm).

In previous studies, some variations of acoustic impedance, such as in the form of linear or quadratic functions, were also calculated with different piezoelectric composites [33,34,35]. In 2008, Zhu Jie et al. used the finite time domain difference method to calculate the spectrum of the gradient change matching layer and found a curve with better ultrasonic transmission property [26]; the result is consistent with the distribution obtained by us. In 2017 and 2021, Li Zheng and Bian Jiacong et al. tried to fabricate a type of gradient matching layer to approach the optimal situation by using tapered high impedance fillers and low impedance substrates [28,39]; this kind of matching layer improves the bandwidth of the transducer to some extent. The best −6 dB bandwidth value of the transducer reaches about 126% (with 1-3 PZT) [39].

Considering the anisotropic feature of the gradient acoustic impedance matching layer, the influences of the matching layer’s thickness on the transducer need to be investigated by simulation once again. Several cases of matching layer thicknesses with the same exponential acoustic impedance distribution mode from 0.1 mm to 0.4 mm are shown in Figure 2c. The corresponding transducer bandwidth is shown in Figure 2d, and the exponential distribution of the acoustic impedance matching layer with a thickness of 0.2 mm (dark green) has the optimized bandwidth. In fact, it has been calculated that the time taken for sound waves to pass through the entire 0.2-mm-thick gradient matching layer is the same as the time taken by a conventional single quarter-wavelength matching layer. The acoustic impedance gradient matching layer also has the same anti-reflection effect. 

## 3. Fabrications and Test of Transducers

### 3.1. Preparation of Acoustic Impedance Gradient Matching Layer

The acoustic impedance matching layer with exponential distribution has been prepared based on the simulation parameters. Tungsten nanoparticles (Aladdin-e) and epoxy resin (Epo-tek 301) were selected as materials. The deposition of tungsten particles in epoxy resin is achieved by a multi-step deposition procedure. We have prepared a variety of mixtures of tungsten particles and epoxy resin in which the filler ratios are different, and then deposit them layer by layer on top of the piezoelectric material under precise control, so that the desired acoustic impedance distribution of the matching layer can be achieved. The flow chart is shown in Figure 3a. Five samples were made in the same condition to reduce the deviation. The appearance of the sample is shown in Figure 3b. The top and bottom surfaces can be observed with a 500 times microscope, shown in Figure 3c,d. It can be seen that the top surface of the sample is uniform epoxy resin, while on the bottom surface the black tungsten powder particles are evenly filled with epoxy resin. Cut the sample lengthwise and observe its cross section with a 100 times microscope to obtain Figure 3e. It can be seen that in the range of 0.2 mm thickness, the filling proportion of tungsten powder gradually increases from the bottom to the top. Under appropriate centrifugal conditions, the acoustic impedance distribution approaches the ideal exponential distribution, as shown in Figure 3h. The acoustic impedance and compression velocity distributions of the matching layer with different volume ratios are shown in Figure 3f. The relationship between volume ratio versus position is shown in Figure 3g.

### 3.2. Fabrication of Transducers

The cutting-filling method was selected to prepare 1-3 piezoelectric composites with an array period of 250 microns and a slit width of 50 microns. The sample was double-sided thinned and polished to the expected thickness, and the ion sputtered 500 nm gold electrode layers. The impedance spectrum of two 1-3 composite piezoelectric materials based on PZT ceramics and PMNT single crystals are shown in Figure 4a,b. The *f*_r_ and *f*_a_ represent resonant and anti-resonant frequencies. The ideal exponential distribution of acoustic impedance matching layers was integrated with 1-3 piezoelectric composites to prepare transducers with a center frequency of 3 MHz. The structural diagram and physical diagram are shown in Figure 4c,d. The ultrasonic transducers were excited by an electrical impulse of 2 μJ and a repetition rate of 1 kHz using an ultrasonic pulse source (5072PR, Olympus). A steel block is placed 3 cm away from the front end of the transducer to reflect the ultrasonic wave. The pulse-echo signal is collected by an oscilloscope (DPO 4104, Tektronix).

### 3.3. Temporal Signal and Frequency Spectrum

The parameters of the different parts of the transducers were listed in Table 2. The temporal signal of each transducer was shown in Figure 5a,c,e. The corresponding spectrum was obtained by Fourier transform, as shown in Figure 5b,d,f. The 1-3 composite PZT transducer with acoustic impedance gradient matching layer has a −6 dB bandwidth of 136.8%, while the single-layer one has 86.6%. By employing 1-3 composite PMN-PT single crystal as the active material, the −6 dB bandwidth of the transducer reaches 174.7%, and the insertion loss is merely −20.3 dB. The transducer achieved a temporal signal close to a single wavelength, which would be beneficial to improve the resolution and quality of the biomedical ultrasound imaging system. The center frequencies of all transducers are very close to the design value (3 MHz). The actual signals of the transducers are consistent with the simulated results. The detailed data are shown in Table 3.

### 3.4. Results Discussion

Some previous transducer bandwidth results are shown in Figure 6a and Table 4. When 1-3 PZT and multiple uniform matching layers are selected, the bandwidth of the transducer cannot exceed 100%. However, when the gradient matching layer is used, the bandwidth of the transducer is greatly improved. When the 1-3 composite single crystal materials are selected, the gradient matching layer makes the bandwidth of the transducer reach over 170%, which completely shows the excellent properties of piezoelectric materials. As can be seen from the schematic diagram of the temporal signal and corresponding bandwidth in Figure 6b, the larger the bandwidth, the smaller the number of periods of the signal and the smaller the trailing of the signal; this will lead to more delicate urban and rural results with higher imaging resolution. So, it is concluded that the increase in the number of matching layers enhances the efficiency of acoustic wave transmission, and therefore, improves the bandwidth of the transducer. The effect of the gradient matching layer on bandwidth improvement is much higher than that of multiple uniform single layers, especially with high-performance single-crystal materials.

### 3.5. Wire Phantom Imaging 

Since the imaging ability of the transducer is closely related to bandwidth, a phantom box with copper wires immersed in water was prepared to take the test. The diameter of each copper wire is 0.5 mm, and the interval between the two wires is 3 cm. A total of three copper wires are arranged at an angle of 45 degrees, and the copper wires are placed perpendicular to the moving direction of the transducer. 

Figure 7a–c show the signal diagram of the wire phantom box obtained by the three transducers mentioned above, respectively. For the amplified signal of the same copper wire, it can be clearly seen that due to the influence of the high-performance single crystal and the superior broadband transmitting capacity of the gradient matching layer, the transducer in Figure 7c obtains the strongest signal, and its axial resolution is much better, which is consistent with the previous experimental results. As for the lateral scale, the three are similar (their −6 dB resolution is about 2.4 mm), indicating that the gradient matching layer does not improve the lateral resolution of the transducer much. Specific resolution results are shown in Table 5. 

## 4. Conclusions

In this work, an anisotropic acoustic impedance gradient composite matching layer for superior broadband ultrasonic transducers has been designed and fabricated by choosing tungsten nano-particles and epoxy resin with proper viscosity. Through a carefully controlled deposit procedure, the volume ratio of the high-impedance filler particles in epoxy resin changes dramatically along the thickness direction, resulting in an ideal acoustic impedance exponential distribution from 8.2 MRayls to 3.2 MRayls within the thickness of 0.2 mm. The experimental results showed that the transducer with this kind of gradient matching layer could exhibit more than 170% of −6 dB bandwidth, which is much higher than that of the one with a single traditional λ/4 matching layer (~80%). In the wire phantom test, the transducer with gradient matching layer also showed much better axial resolution (~0.3 mm for 1-3 PMN-PT at 3 MHz); the transducer achieved a temporal signal close to a single wavelength, thus there is the potential to greatly improve the resolution and imaging quality of the biomedical ultrasound imaging system.

## Figures and Tables

**Figure 1 sensors-22-08025-f001:**
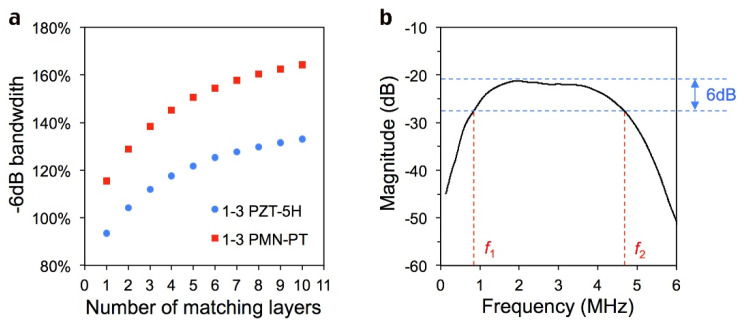
(**a**), −6 dB bandwidth of transducers with different number of matching layers. (**b**), Diagram of a transducer’s spectrum.

**Figure 2 sensors-22-08025-f002:**
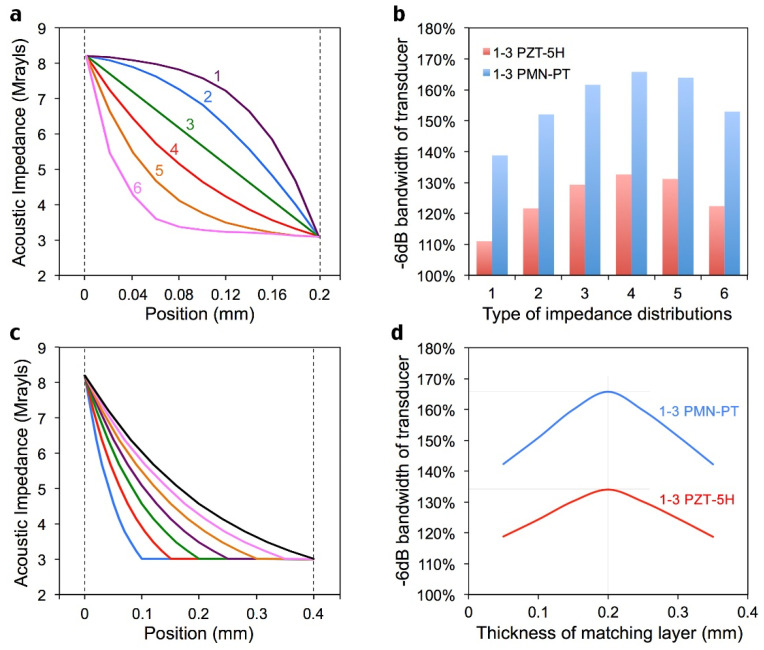
Simulation results of different gradient matching layers. (**a**), Different types of acoustic impedance distribution. (**b**), Corresponding −6 dB bandwidth of various cases in (**a**). (**c**), Gradient matching layer with different thickness. (**d**), Corresponding −6 dB bandwidth of various cases in (**c**).

**Figure 3 sensors-22-08025-f003:**
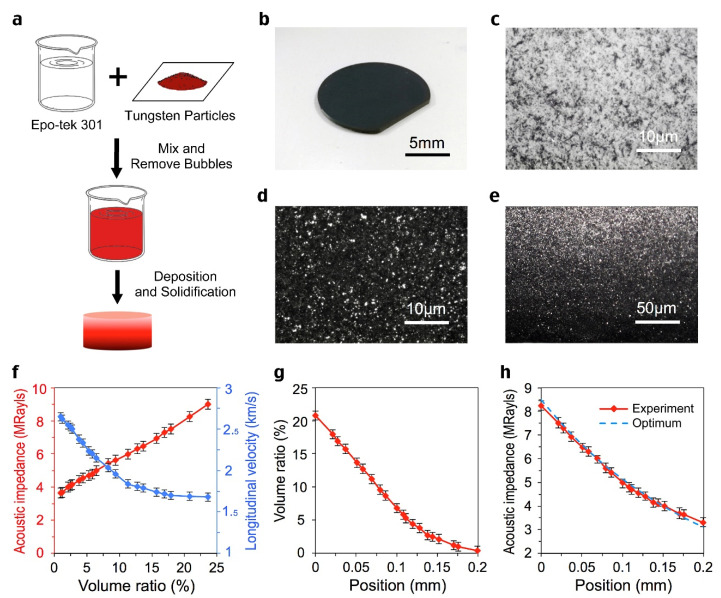
**Fabrication procedure and acoustic properties of gradient matching layer.** (**a**), Schematic diagram of the manufacturing steps of the matching layer. (**b**), The appearance of the matching layer sample. (**c**), The top surface morphology (500× magnification). (**d**), The bottom surface morphology (500× magnification). (**e**), The cross-section morphology (100× magnification). (**f**), Acoustic impedance and Longitudinal wave velocity of matching layers with different volume ratios. (**g**), The volume ratios of filler at various positions. (**h**), The testing acoustic impedance distribution of matching layer. The thickness is 0.2 mm, and the acoustic impedance decreases from 8.2 to 3.2 MRayls. The curve is very close to the optimal one.

**Figure 4 sensors-22-08025-f004:**
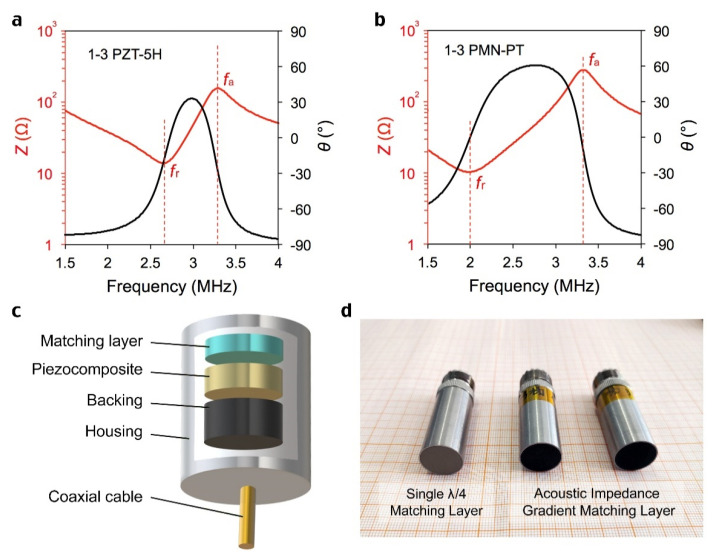
(**a**), Impedance spectrum of 1-3 composite PZT-5H ceramics. (**b**), Impedance spectrum of 1-3 composite PMN-PT single crystals. (**c**), Structure diagram of transducer. (**d**), Appearance diagram of transducers. The filler of the quarter-wavelength matching layer is SiC particles (grey), while the filler of the gradient matching layer is tungsten particles (black).

**Figure 5 sensors-22-08025-f005:**
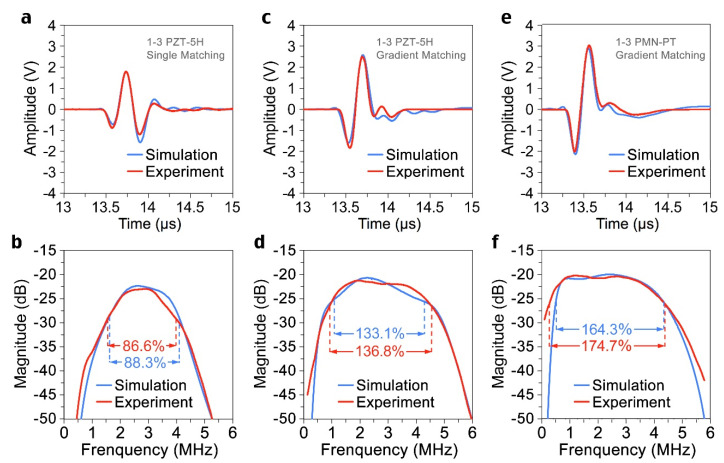
**Simulated and experimental results of transducers.** First echo signals were shown of transducers with (**a**) single λ/4 matching layer and 1-3 composite PZT-5H (**c**) acoustic impedance gradient matching layer and 1-3 composite PZT-5H and (**e**) acoustic impedance gradient matching layer and 1-3 composite PMN-PT. The corresponding spectrums were shown of transducers with (**b**) single λ/4 matching layer and 1-3 composite PZT-5H (**d**) acoustic impedance gradient matching layer and 1-3 composite PZT-5H and (**f**) acoustic impedance gradient matching layer and 1-3 composite PMN-PT.

**Figure 6 sensors-22-08025-f006:**
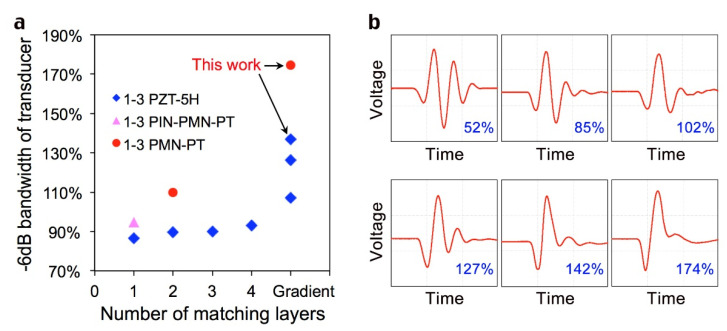
(**a**), −6 dB bandwidth of transducers with different number of matching layers. (**b**), Temporal signals and corresponding bandwidths.

**Figure 7 sensors-22-08025-f007:**
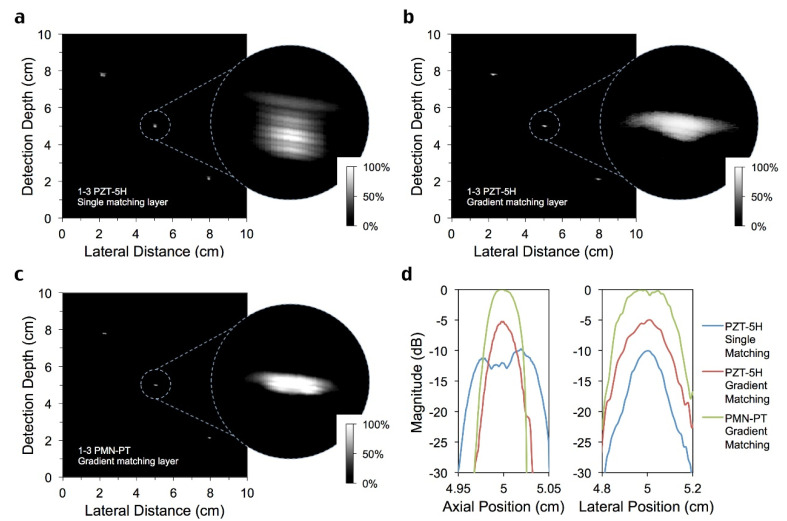
**Imaging test results of transducer with** (**a**) single λ/4 matching layer and 1-3 composite PZT-5H, (**b**) acoustic impedance gradient matching layer and 1-3 composite PZT-5H and (**c**) acoustic impedance gradient matching layer and 1-3 composite PMN-PT. (**d**) Axial and lateral resolution of transducers.

**Table 1 sensors-22-08025-t001:** Acoustic parameters of 1-3 composite piezoelectric materials.

Parameter	1-3 Composite PZT-5H	1-3 Composite PMN-PT
*Area*	78.54 mm^2^ (radius = 5 mm)	78.54 mm^2^ (radius = 5 mm)
*Thickness*	0.56 mm	0.56 mm
*Density*	4052 kg/m^3^	4086 kg/m^3^
*Longitudinal wave velocity*	3912 m/s	3893 m/s
*d* _33_	~610 pC/N	~1230 pC/N
*k* _t_	0.58	0.81
$\varepsilon_{33}^{S}$	1211 $\varepsilon_{0}$	822 $\varepsilon_{0}$
$\delta_{e}$	0.037	0.030
$\delta_{m}$	0.018	0.015

**Table 2 sensors-22-08025-t002:** Parameters of different parts of transducers.

Material	Thickness (mm)	Acoustic Impedance (MRayls)	Longitudinal Wave Velocity (m/s)	Attenuation Coefficient at 3 MHz (dB/cm)
1-3 composite PZT-5H	0.56	15.8	3912	-
1-3 composite PMN-PT	0.56	15.9	3893	-
Matching layer 1	0.02	8.2	1700	2.0
Matching layer 2	0.02	7.2	1740	2.2
Matching layer 3	0.02	6.3	1810	2.4
Matching layer 4	0.02	5.6	1900	2.7
Matching layer 5	0.02	4.9	2100	3.0
Matching layer 6	0.02	4.4	2300	2.5
Matching layer 7	0.02	4.0	2500	2.2
Matching layer 8	0.02	3.6	2550	1.8
Matching layer 9	0.02	3.3	2600	1.4
Matching layer 10	0.02	3.2	2650	1.0
Single matching layer	0.20	5.5	2800	1.0
Backing	30	10.5	1500	~10

**Table 3 sensors-22-08025-t003:** Measured performance parameters of transducers.

Performance	Single-Matching-Layer Transducer (PZT-5H)	Gradient-Matching-Layer Transducer (PZT-5H)	Gradient-Matching-Layer Transducer (PMN-PT)
Center frequency	3.02 MHz	2.98 MHz	3.06 MHz
Insertion loss	−23.1 dB	−21.3 dB	−20.3 dB
−6 dB bandwidth	86.6%	136.8%	174.7%

**Table 4 sensors-22-08025-t004:** The −6 dB bandwidth of transducers with different matching layers.

Piezoelectric Material	Number of Matching Layers	−6 dB Bandwidth	Reference
1-3 PZT-5H	1	86.6%	This work
1-3 PZT-5H	2	89.8%	[40]
1-3 PZT-5H	3	90.0%	[14]
1-3 PZT-5H	4	93.0%	[41]
1-3 PZT-5H	Gradient	107.0%	[28]
1-3 PZT-5H	Gradient	126.3%	[39]
1-3 PZT-5H	Gradient	136.8%	This work
1-3 PIN-PMN-PT	1	94.6%	[42]
1-3 PMN-PT	2	110.0%	[22]
1-3 PMN-PT	Gradient	174.7%	This work

**Table 5 sensors-22-08025-t005:** The −6 dB resolutions of transducers.

Material	Axial Resolution	Lateral Resolution
1-3 PZT-5H with single matching layer	0.82 mm	2.3 mm
1-3 PZT-5H with gradient matching layer	0.31 mm	2.5 mm
1-3 PMN-PT with gradient matching layer	0.30 mm	2.4 mm

## Data Availability

The data presented in this study are available on request from the corresponding author.

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
