# Peer review of "Enhancement of Ultrasonic Transducer Bandwidth by Acoustic Impedance Gradient Matching Layer"

_sensors, 2022, doi:10.3390/s22208025_

Round 1
Reviewer 1 Report
As the opinion of the reviewer, the following improvement should be made.
1. Error in Table 1: radio;
2. Parameter t is used with two meanings: The thickness of piezoelectric material is shown in Table 1; matching layer thickness in line 112 and figure 1(d)-(e);
3. Please explain the rationality of choosing 10 single layers or the relationship between the number of layers and the convergence of simulation;
4. Please explain reference point of -6 dB bandwidth, in other words, specify to whom the loss of -6 dB is compared;
5. The figure 1 shows the results of four new gradient impedance materials. Please add reference materials for comparison and analysis;
6. A curve can be fitted by many types of functions, and four distribution patterns are better expressed by mathematical expressions;
7. For (e) in Figure 1, please explain how to determine that t=0.2 mm is optimal;
8. “the volume ratio of the high-impedance filler powder” is described in line 75 on page 2, and “the density of tungsten particles” is expressed in the conclusion. Please give a definite description;
9. Please explain the meaning of fa and fr in Figure 3 (a) and (b);
10. Relationships of Z- volume ratio and Z- position are given in Figure 2. Please add the relationship of volume ratio versus position.
Author Response
Dear Editor and Reviewers, We really appreciate your valuable advice and also the suggestions from reviewers, which have greatly helped us improve the overall quality of the manuscript. Some latest references have been added. The results acquired in these articles were also discussed and compared in our revised manuscript. Some spelling mistakes in the text were corrected. The response to reviewer point to point are listed in the attachment, shown in red and blue color. We are looking forward to your comment and decision soon. Yours sincerely, Rui Zhang October 11, 2022

Reviewer 2 Report
The authors describe in this article the simulation and the fabrication of graded impedance matching layer to enhance piezo-based transducers' functionality.
Even if the subject is very interesting and is clearly useful for the community, I find that this article has several flaws.
Firstly, this subject is currently active and most of the references are pretty old. I think that it should be interesting to add several references to show how active this subject is.
Secondly, I think that the English language level should be improved to meet scientific standards. Some typo mistakes in the text should also be removed.
simulation :
The simulation part of the article is very close to other articles and J. Zhu's thesis. The authors explain the simulation for PZT composite even if they study two piezo composites. The authors should maybe explain why they choose only 1-3 composites even if this method can be used for crystals.
The results show that exponential gives the best response, as already known in the literature. Authors could maybe precise it in their comments. The notion of "ideal" seems obscure to me. Do they optimize the parameters? if Yes, explain how.
The authors said that the concept of wavelength is not valid in anisotropic media. They should say that the concept of a quarter wavelength matching layer is not relevant, the wavelength still being a concept in anisotropic media.
The authors said that they take 4 distinct thicknesses. an optimisation should be done on this parameter instead of only 4 discrete values.
Fabrication :
I am not an expert on this part and my comments are just for information.
I personally don't understand how the authors can change the distribution of the tungsten pellet in their matching layer, which seems to be done only by deposition. In this case, the decay should be the same whatever the thicknesses of the fabricated matching layer and it seems complex to fit with the curve given in figure 1d.
The results parts for temporal signal/spectrum and imaging should be enhanced and a comparison with previous results from the literature should be added. Without that, it is difficult to measure the impact of the proposed graded impedance layer.
The conclusion should also be improved.
Author Response
Dear Editor and Reviewers, We really appreciate your valuable advice and also the suggestions from reviewers, which have greatly helped us improve the overall quality of the manuscript. Some latest references have already been added to replace the old ones. The results acquired in these articles were also discussed and compared in our revised manuscript. Some typo mistakes in the text were removed. The response to reviewer point to point are listed in the attachment, shown in red color. We are looking forward to your comment and decision soon. Yours sincerely, Rui Zhang October 11, 2022

Round 2
Reviewer 2 Report
Dear authors,
Thanks for the consideration you had for my remarks.